# Screening and diversity analysis of Dibutyl phthalate degrading bacteria in agricultural soil in Chengdu, China

**Yong Li** [ORCID] *, **Sidan Gong, Huan Liu, Yajie Li**

School of Environmental Science and Engineering, Southwest Jiaotong University, Chengdu, China

* liyong@swjtu.edu.cn

## Abstract

Dibutyl phthalate (DBP) with teratogenicity, carcinogenesis, and mutagenesis, is a ubiquitous endocrine disruptor in the environment. The widespread usage of plastic mulch has resulted in a severe DBP pollution problem in agricultural soil. One of the most vital ways to mitigate the DBP pollution problem is to use DBP degrading bacteria to reduce the concentration of DBP in agricultural soil. DBP degrading bacteria studied in previous studies mainly come from sewage treatment plants, rivers, landfills, bioreactors, and other environmental media. At the same time, there was relatively little research on DBP degrading bacteria in agricultural soil. Therefore, using the pure culture approach, the candidate DBP degrading strains were isolated and screened from vegetable plots contaminated with plastic mulch in Dayi County, Chengdu City, Sichuan Province, China. The taxonomy of the strains was determined using the 16 sedimentation ribosomal deoxyribonucleic acid (16S rDNA) molecular technique. Furthermore, ultraviolet and visible spectrophotometry and high performance liquid chromatography (HPLC) were used to assess the degrading capability of strains. The findings showed that a total of twenty-six DBP degrading strains were screened from vegetable plots contaminated with plastic mulch in Dayi County, Chengdu City, Sichuan Province, China, and these strains belonged to two phyla: *Proteobacteria* and *Bacillota*, eight families, including *Pseudomonas*, *Enterobacteriaceae*, and *Bacillaceae*, as well as ten genera, such as *Enterobacter*, *Pseudomonas*, and *Bacillus*. One of them, the strain SWDB-7 was a potentially new species in the *Enterobacter* genus. The most prominent genus of the isolated strains was *Enterobacter*. There were significant variations in the degradation capability of different obtained DBP degrading strains. and their degradation efficiency was 14.12%-91.86%. With a total DBP removal rate of 91.86%, strain SWDB-15 had the most effective degrading capability among them. To sum up, the DBP degrading strains in vegetable plots contaminated with plastic mulch in Dayi County, Chengdu City, Sichuan Province, China are rich in diversity and capable of breaking down DBP.

**Data Availability Statement:** All Accession Number files are available from the National Center for Biotechnology Information (NCBI) database (https://www.ncbi.nlm.nih.gov/) (accession number(s) (ON428492, ON428524,PP348297,

ON428982,ON428983,ON545724,ON428491,
ON428981,PP348298,PP348300,ON428989,
ON428987,ON428988,ON428991,ON428523,
ON428980,PP348299,ON428984,ON428985,
ON428986,ON428990,ON545723,ON545722,
ON428992,PP348301.).

**Funding:** The author(s) received no specific
funding for this work.

**Competing interests:** The authors have declared
that no competing interests exist.

# 1 Introduction

Dibutyl phthalate (DBP) is a typical environmental endocrine disruptor, which is utilized extensively as a plasticizer in the plastic manufacturing industry [1]. Because of its teratogenic, carcinogenic, and mutagenic toxicity to organisms, it has garnered a lot of attention [2]. In 1977, the U.S. Environmental Protection Agency (EPA) included DBP in the list of 129 pollutants under key control, and China also listed DBP on a blacklist of priority pollutants in 2002 [3]. Agricultural soil is a considerable portion of the country's production base for goods including grains, cotton, oil and sugar vegetables, and fruits in China, and its environmental quality is closely bound up with human health. Due to its high Octanol-Water Partition Coefficient, DBP is readily adsorbed into soil particles, which makes agricultural soil a significant "sink" of DBP in the environment [4]. Lu et al. [5] summarized the current situation of DBP content in the agricultural soil of Denmark, the Netherlands, the United Kingdom, and the provinces of East, South, Northeast, and North China. They discovered that the evaluated DBP content in the agricultural soil of Denmark, Netherlands, and the United Kingdom was 0.11, 0.006, and 0.0008 mg/kg respectively. The DBP content in agricultural soil in China ranges from 0.001 mg/kg to 1000 mg/kg. Moreover, the DBP content in the soil of different provinces, different cities, and even different functional areas of the same city in China varies significantly. To date, apart from China, relatively few studies have been conducted on the DBP content in agricultural soil in other countries worldwide. This might be attributed to the fact that in other countries, especially in Western ones, less and standardized plastic mulch is utilized in agricultural production, and the application of chemical fertilizers and pesticides is also less. Thus the DBP pollution in agricultural soil is not severe. However, the usage of plastic films, chemical fertilizers, and pesticides in farmland soil in China contrasts with that in Western countries, leading to an extremely serious DBP pollution situation in farmland soil in China [6]. Currently, the average content of DBP in agricultural soil across the countries exceed the soil PAEs compounds control standard in the United States [7]. Thus, it has attracted the interest of both local and federal agencies. Based on the research findings of Kong et al. [8], the presence of higher levels of human intervention and chemical substances such as fertilizers, pesticides, and agricultural mulching film in vegetable fields results in a higher DBP content compared to other types of farmland soils such as rice field soil and orchard soil.

Plastic mulch is one of the essential sources of DBP in agricultural soil [9]. To maintain the soil temperature and moisture, inhibit the growth of weeds, weaken the harm of diseases and pests, shorten the growth cycle of crops, and increase the yield of crops, mulch cultivation technology is adopted in the cultivation process. According to the National Bureau of Statistics, in 2022, the amount of plastic mulches used in the country was 1.342 million tons, and the plastic mulches covered an area of 17,470.9 thousand hectares [10]. As a result of farmers' limited awareness of recycling, outdated recycling methods, and an imperfect recycling system, a large amount of plastic mulches was left behind [11]. DBP is easy to seep from the plastic mulches under the action of rain leaching, weathering, and ultraviolet radiation [12]. It damages soil quality [13] and affects soil nutrient content [14]. It can also enter the food chain from the soil and accumulate along the food chain, ultimately endangering human health [15]. Currently, known DBP treatment techniques are divided into abiotic degradation (including physical treatment and chemical treatment) and biodegradation (including microbial degradation and phytoremediation). Among them, microbial degradation stands out by virtues of its high efficiency, cost-effective, safe, stable, mature, and no secondary pollution, and it has become the most suitable means to remove DBP in agricultural soil [16]. Additionally, plastic mulch, regarded as a microplastic pollutant, not only fails to exert a significant impact on the diversity of soil microorganisms but also can significantly enhance the quantity of soil

microorganisms, thereby rendering biodegradation more feasible [17]. Therefore, it is of great research significance to screen out DBP degrading strains with strong adaptability and degradation capability from agricultural soil. It can serve as a new bacterial resource for controlling DBP pollution in agricultural soil.

At present, many scholars have reported on DBP degrading bacteria, which mainly come from environmental media such as sewage treatment plants, rivers, landfills, and bioreactors. For instance, Jin et al. [18] isolated a bacterial strain of *Gordonia* that can degrade DBP from activated sludge; from leachate samples of municipal solid waste, Kumar et al. [19] newly separated and screened two DBP degrading bacterial strains belonging to *Pseudomonas* and *Comamonas* respectively; Prasad et al. [20] isolated a DBP degrading bacterium of *Variovorax* genus from garbage dumped soil by enrichment technique. While there is a relative lack of research on DBP degrading bacteria in agricultural soil [4, 21–23]. Duan et al. [24] isolated and selected two strains belonging to the *Acinetobacter* genus and *Pseudomonas* genus from the vegetable field of South China Agricultural University. By utilizing DBP as the sole carbon and energy source, the degradation efficiency of DBP reached 98.64% and 74.62%, respectively, within a span of 2 days at an initial concentration of 40 mg/L. Pan et al. [25] isolated a strain of DBP degrading bacteria belonging to the *Sphingobium* genus from vegetable fields in Qingdao and analyzed its degradation performance. The results showed that when the initial concentration of DBP was 100 mg/L, the degradation efficiency of the strain was 71.43% in 5 days. A DBP degrading strain of the *Geotrichum* genus was isolated from the soil layer of a rice field in a city in western Guangdong Province by Wang et al. [26] When the initial concentration of DBP in the cultivated soil was 50 mg/L, the degradation efficiency of this strain could reach 84.13% within 7 days. Kong et al. [27] isolated a DBP degrading bacterium from the soil of the simulated agricultural ecosystem in Haidian District, Beijing, belonging to the *Gordonia* genus. It can accelerate the degradation of DBP in agricultural soil and reduce the accumulation of DBP in plants. In summary, there are few DBP degrading strains isolated at present, and their degradation capability is poor. Therefore, the research on the diversity and degradation efficiency of DBP degrading bacteria in agricultural soil is of great significance, which can enrich bacterial species resources and provide a theoretical basis for DBP management in agricultural soil.

According to the "China Rural Statistical Yearbook" issued by the Department of Rural Social and Economic Survey of the National Bureau of Statistics, the use and coverage area of plastic mulch in Sichuan Province often ranks among the top in the whole country. According to the investigation of the Sichuan Provincial Department of Ecology and Environment, at present, about 20% of the plastic mulch in Sichuan Province has not been recycled, resulting in a large number of mulch residues remaining in the agricultural soil. In addition, the use of plastic mulch on vegetable soil is much higher than that on other agricultural soil, leading to a more serious DBP pollution problem [8]. The aim of this study was to screen out a variety of highly effective DBP degrading bacteria from the vegetable plots contaminated with plastic mulch, to provide rich bacterial resources and a theoretical basis for the control of DBP pollution in agricultural soil.

It was hypothesized that a rich diversity of DBP degrading bacteria could be screened from vegetable plots contaminated with plastic mulch in Dayi County, Chengdu City, Sichuan Province, China, and the strains possess superb DBP degradation capability. Therefore, in this study, the pure culture method was used to isolate and screen the candidate DBP degrading strains. Soil samples were collected from vegetable plots contaminated with plastic mulch in Dayi County, Chengdu City, Sichuan Province, China. After enrichment culture in liquid medium with DBP as the only carbon source and energy source, the colonies with transparent circles and colonies with clear morphology and good growth but without transparent circle were separated and purified by the dilution coating plate method and quadrant streaking

method. The degradation efficiency of the strain was qualitatively and quantitatively analyzed by Ultraviolet and visible spectrophotometry and high performance liquid chromatography (HPLC), respectively. The determination of the degradation efficiency was conducted to ascertain whether the strain is a DBP degrading bacterium. The 16 sedimentation ribosome deoxyribonucleic acid (16S rDNA) molecular technique was used to identify the strains, determine the taxonomic status of the strains, and explore their biodiversity.

## 2 Materials and methods

### 2.1 Sampling description

Three soil samples were collected from rural area of Dayi County, Chengdu City, Sichuan Province, China (The coordinates of this place are E103.468897° and N 30.555390°) under stable weather conditions in December of 2022. All samples were collected for vegetable soils (the vegetables including cauliflower, cabbage and kohlrabi). Sampling sites were selected at plots contaminated with plastic mulch.

Soils (0–20 cm depth) were collected using a pre-cleaned stainless spade into pre-cleaned sterile polyethylene bags. If necessary, particles of roots and leaves of vegetables were removed before sampling. The samples were cooled in a foam box with ice cubes during transport to the laboratory. During sampling, precautions were taken to avoid DBP contamination. Then the samples were air-dried and freeze-dried, ground, homogenized by sieving through a 60-mesh stainless steel mesh. After sufficient mixing, homogenized soil samples were stored in sterile polyethylene bags at -20°C until extraction.

### 2.2 Chemicals, reagents and media

In this study, the necessary chemicals, reagents and media were prepared according to the method of Lu [28].

DBP (chromatographic grade) was purchased from Sigma-Aldrich (Saint Louis, Missouri, USA). Methyl alcohol (chromatographic grade) was purchased from EMD Millipore Corporation (Shanghai, China). The mother liquor was configured with methyl alcohol and DBP according to the experimental requirements. Formylic acid (chromatographic grade) was purchased from Macklin Biochemical CO., Ltd (Shanghai, China). All the other chemical reagents in the experiment (analytically pure) were purchased from Sinopharm Chemical Reagent Co., LTD. (Shanghai, China).

Minimal salt liquid medium used was composed of NaCl, 0.75 g; $NH_4NO_3$, 0.5 g; $FeCl_3$, 0.001 g; $K_2HPO_4 \cdot 3H_2O$, 1 g; $MgSO_4 \cdot 7H_2O$, 0.4 g; $CaCl_2$, 1.9 g; and distilled water, 1000 mL; with a pH 7.5–8.0. The DBP mother liquor was added according to the specific requirements of the experiment. Minimal salt solid medium was supplemented with an additional 20 g AGAR.

The Luria-Bertani (LB) medium used was composed of NaCl, 10 g; yeast paste, 5 g; peptone, 10 g; and distilled water, 1000 mL; with a pH 7.0–7.2. LB solid medium required additional 20 g agar.

### 2.3 Isolation and screening of DBP degrading strains

Each 5.0 g air-dried soil sample was added into a 250 ml conical flask containing 100 mL minimal salt liquid media (containing DBP 0.1mmol /L). The conical flask was placed in a constant temperature shaking incubator for Enrichment cultivation, under the condition of 28°C, 180 rpm for 7 days. This was followed by transfer to 100 mL freshly prepared minimal salt liquid media at 5% inoculations each time (At the same time, the concentration of DBP in the media

was gradually increased to 0.2, 0.3, 0.4, 0.5 mmol/L in sequence.). And they were cultured under the same conditions. At the end of the enrichment culture, a stable flocculation will be formed in the conical flask, that is the enriched solution with the capability to stably degrade DBP. $10^{-1}$ bacterial suspension was prepared by adding 1 mL of DBP enrichment solution (0.5 mmol/L) to 9 mL of aseptic distilled water. The bacterial suspension was taken and diluted step by step to $10^{-2}$–$10^{-6}$. 0.1 mL of each bacterial suspension was applied on minimal salt solid media (containing 0.5 mmol/L DBP). These inoculated minimal salt solid media were placed in a constant temperature incubator at 28˚C for 2 days. Next, the colonies with transparent circles or colonies without transparent circle but with clear morphology and good growth were picked, separated, and purified for more than four rounds. After the single colony was confirmed as pure culture by morphological characteristics and microscopic examination, the LB slant medium was inoculated, cultured and stored in a 4˚C refrigerator for further use.

## 2.4 Molecular biological identification of DBP degrading bacteria

**2.4.1 Genomic DNA Extraction and PCR amplification of 16S rDNA.** The total bacterial DNA was extracted with the bacterial genomic DNA extraction kit from Sangon Biotech (Shanghai) Co., Ltd, following the kit instructions. The universal primers 27F (5'-AGAGTT TGATCCTGGCTCAG-3') and 1492R (5' -GGTTACCTTGTTACGACTT-3') were used for the PCR amplification of 16S rDNA [29]. The PCR was carried out in a BioRad T100 Thermal Cycler. The PCR reaction system (20 μL) was: 10 × Ex Taq buffer, 2.0 μL; 5 U Ex Taq, 0.2 μL; 2.5 mmol/L dNTP Mix, 1.6 μL; 27F, 1 μL; 1492R, 1 μL; DNA, 0.5 μL; and ddH$_2$O, 13.7 μL. The PCR was carried out as follows: Initial denaturation at 95˚C for 5 min, 25 cycles of denaturation at 95˚C for 30 s, annealing at 56˚C for 30 s, extension at 72˚C for 90 s, and with a final extension at 72˚C for 10 min. The PCR amplification products were detected by 0.8% agarose gel electrophoresis. Then PCR products were purified by the kit and sent for sequencing to Sangon Biotech (Shanghai) Co., Ltd by 3730XL sequencer.

**2.4.2 Phylogenetic analysis.** Using SeqMan II (version 15.90, DNASTAR, Madison, WI, USA, 2018), the contaminated sequences from the raw bacterial 16S rDNA sequence data were removed. Furthermore, the obtained sequences were curated to remove regions with low-quality nucleotide scores (i.e., ambiguous bases) and assembled into contigs. The obtained 16S rDNA full sequences were sent for the Basic Local Alignment Search Tool (BLAST) analysis at EzBioCloud (https://www.ezbiocloud.net) to find the closest species with the highest similarity to the strains. Then, similar and effectively published sequences of typical strains were retrieved from the database. After conducting multiple sequence alignment using clustalX (version 2.1, EMBL, Heidelberg, Germany, 2012), sequence editing was carried out in MEGA X (version 10.05, Institute for Genomics and Evolutionary Medicine, Temple University, Philadelphia, PA, USA, 2018) to estimate the evolutionary distance utilizing the Kimura 2-parameter (K2P) model. Finally, the 16S rDNA phylogenetic tree was constructed using the Neighbor-Joining (NJ) method with a self-expanding value of 1000 [30].

## 2.5 Study on the degradation characteristics of DBP degrading strains

**2.5.1 Ultraviolet and visible spectrophotometry determines degradation capability.** The degradation capability of DBP degrading bacteria was qualitatively determined by ultraviolet and visible spectrophotometry and the quality assurance (QA) / quality control (QC) employed is conducted in accordance with the study of Lu [28]. The minimum detectablity of DBP in minimal salt liquid medium was 0.01 mmol/L. The recovery efficiency of DBP ranged from 91.6% to 96.4%. The relative standard deviation (RSD) ranging from 0.22% to 0.56% (n = 3). Each isolated colony was inoculated in a 250 mL conical flask containing 100 mL of LB

liquid media. These conical flasks were placed in a constant temperature shaking incubator under the condition of 28˚C, 180 rpm for 48 hours. After centrifuging at 4000 rpm for 5 minutes, the supernatant was discarded, and distilled water was added and mixed with a vortex to prepare seed fermentation broth with an $OD_{600}$ (Optical Density) value of approximately 1.0. 2 mL of seed fermentation broth with $OD_{600}$ of approximately 1.0 was inoculated into 100 mL of minimal salt liquid medium (containing DBP 0.5 mmol/L). At the same time, for each sample, three replicates were set up, and the media with no cells inoculated was set up as control. Samples and replicates were placed in a constant temperature shaking incubator under the condition of 28˚C, 180 rpm for 48 hours to acquire bacterial fermentation broth. The bacterial fermentation broth was added to a 250 mL conical flask filled with 50 mL methyl alcohol, and the conical flask was shaken for 1 min to terminate the reaction, and then centrifuged at 4000 rpm for 5 min. After that, ultraviolet and visible spectrophotometry was used to scan in the wavelength range of 200–350 nm to determine the maximum absorption wavelength of each bacterial fermentation broth. According to the difference of absorbance at the maximum absorption wavelength between the bacterial fermentation broth and the replicate, the degradation capability of the strain was preliminarily determined.

**2.5.2 Determination of degradation efficiency by HPLC.** External standard method [31] was used to quantify DBP. First, a standard curve was constructed (The R-squared ($R^2$) of the standard curve is above 0.999). Then according to the detection results of ultraviolet and visible spectrophotometry, the target strain with degradation effect was prepared into a seed fermentation broth with $OD_{600}$ of about 1.0, the preparation method was the same as 2.5.1. 2 mL of seed fermentation broth with $OD_{600}$ of about 1.0 was prepared into bacterial fermentation broth and replicates were set, the method was also the same as 2.5.1. The method of DBP detection by HPLC is mentioned in Lu's [28] article. The bacterial fermentation broth was added into a 250 mL conical flask filled with 102 mL methyl alcohol, and the conical flask was shaken for 1 min to terminate the reaction. After centrifuging at 4000 rpm for 5 minutes, 1 mL of supernatant was taken and filtered by 0.22μm nylon filter membrane. The filtrate was dissolved by ultrasound with a span of 30 minutes for quantitative detection by liquid chromatography. The chromatographic conditions were as follows: The chromatographic column was C18 reversed phase InertSustain (4.6 mm×150 mm×5 μm). The mobile phase is the volume ratio of methyl alcohol to water (containing 0.1% formic acid) is 90:10; The flow rate was 0.8 mL·min$^{-1}$. The detection wavelength was 230 nm. The column temperature was 30˚C. The sample size was 20 μL.

## 2.6 Statistical analysis

The data from ultraviolet and visible spectrophotometry was analyzed by Origin software (version 9.1, OriginLab, Northampton, Massachusetts, USA, 2013). The data from HPLC was analyzed using LabSolutions CS software (version 6.108, Shimadzu (China) CO., LTD., Shanghai, China, 2021). Statistical analysis was performed using SPSS statistical software (version 20.0, IBM, Armonk, NY, USA, 2013). Significance levels were using $P = 0.05$ in all statistical analyses.

## 3 Results

### 3.1 Results of isolation and screening

DBP degrading strains were isolated and screened by the dilution coating plate method and quadrant streaking method. Combined with the characteristics of colony growth, morphology, and transparent circle, 32 strains of single bacteria were isolated from a vegetable field polluted by plastic mulch in Dayi County, Chengdu City, Sichuan Province, China. After qualitative

**Table 1. List of whether the 26 DBP degrading bacteria strains could form transparent circles on minimal salt solid medium containing 0.5 mmol/L DBP.**

| Bacteria Strain | Transparent circle | Bacteria Strain | Transparent circle | Bacteria Strain | Transparent circle |
|---|---|---|---|---|---|
| SWDB-1 | + | SWDB-2 | + | SWDB-3 | - |
| SWDB-4 | + | SWDB-5 | + | SWDB-6 | + |
| SWDB-7 | + | SWDB-8 | + | SWDB-9 | + |
| SWDB-10 | - | SWDB-11 | + | SWDB-12 | - |
| SWDB-13 | + | SWDB-14 | + | SWDB-15 | + |
| SWDB-16 | - | SWDB-17 | + | SWDB-18 | + |
| SWDB-19 | - | SWDB-20 | + | SWDB-21 | + |
| SWDB-22 | + | SWDB-23 | + | SWDB-24 | + |
| SWDB-25 | + | SWDB-26 | + | | |

Note: "+" presents that the strain can form transparent circles on minimal salt solid medium containing 0.5 mmol/L DBP. "-" presents that the strain can't form transparent circles on minimal salt solid medium containing 0.5 mmol/L DBP.

and quantitative detection, a total of 26 strains of DBP degrading bacteria with stable degradation effects were obtained. Among them, 21 strains formed transparent circles on minimal salt solid medium containing 0.5 mmol/L DBP, and 5 strains did not form transparent circles, namely strains SWDB-3, SWDB-10, SWDB-12, SWDB-16 and SWDB-19, as shown in Table 1. The images of transparent circles on minimal salt solid medium containing 0.5 mmol/L DBP are presented in S1 Fig.

## 3.2 Strain identification and diversity analysis

The DNA of 26 strains of DBP degrading bacteria were sequenced, and the sequencing results were submitted to the National Center for Biotechnology Information (NCBI) database (https://www.ncbi.nlm.nih.gov/) for Blast homology analysis, as shown in S1 Table. The phylogenetic evolutionary tree is shown in Fig 1.

According to the results of 16S rRNA gene sequence analysis, the 26 isolated strains were classified into two phyla: *Proteobacteria* and *Bacillota*; Four classes: *Alphaproteobacteria*, *Betaproteobacteria*, *Gammaproteobacteria* and *Bacilli*; Five orders: *Enterobacteriales*, *Pseudomonadales*, *Bacillales*, *Hyphomicrobiales* and *Burkholderias*; Eight families: *Pseudomonas*, *Enterobacteriaceae*, *Bacillaceae*, *Comamonadaceae*, *Alcaligenaceae*, *Erwiniaceae*, *Sphaerotilaceae* and *Rhizobiaceae*; Ten genera: *Enterobacter*, *Pseudomonas*, *Bacillus*, *Variovorax*, *Achromobacter*, *Agrobacterium*, *Delftia*, *Ideonella*, *Kluyvera* and *Phytobacter*. Among them, *Enterobacter* DBP degrading bacteria were the most numerous, with a total of six strains. It was followed by *Pseudomonas*, *Bacillus*, *Variovorax*, *Achromobacter*, and *Agrobacterium*, with four strains, three strains, three strains, two strains, and two strains of DBP degrading bacteria. The other genera had only one DBP degrading bacteria each.

## 3.3 Analyses of degradation characteristics of DBP degrading strains

**3.3.1 Determination of degradation capability by ultraviolet and visible spectrophotometry.** The data from the ultraviolet and visible spectrophotometry was analyzed using Origin software. The results showed that the maximum absorption wavelength of each bacterial fermentation broth was about 230 nm. The absorbance of the control at the maximum absorption wavelength is 1.75. The absorbance of each fermentation broth at the maximum absorption wavelength was significantly lower than that of the replicate. These results indicated that twenty-six strains of DBP degrading bacteria were able to degrade part of DBP within 48

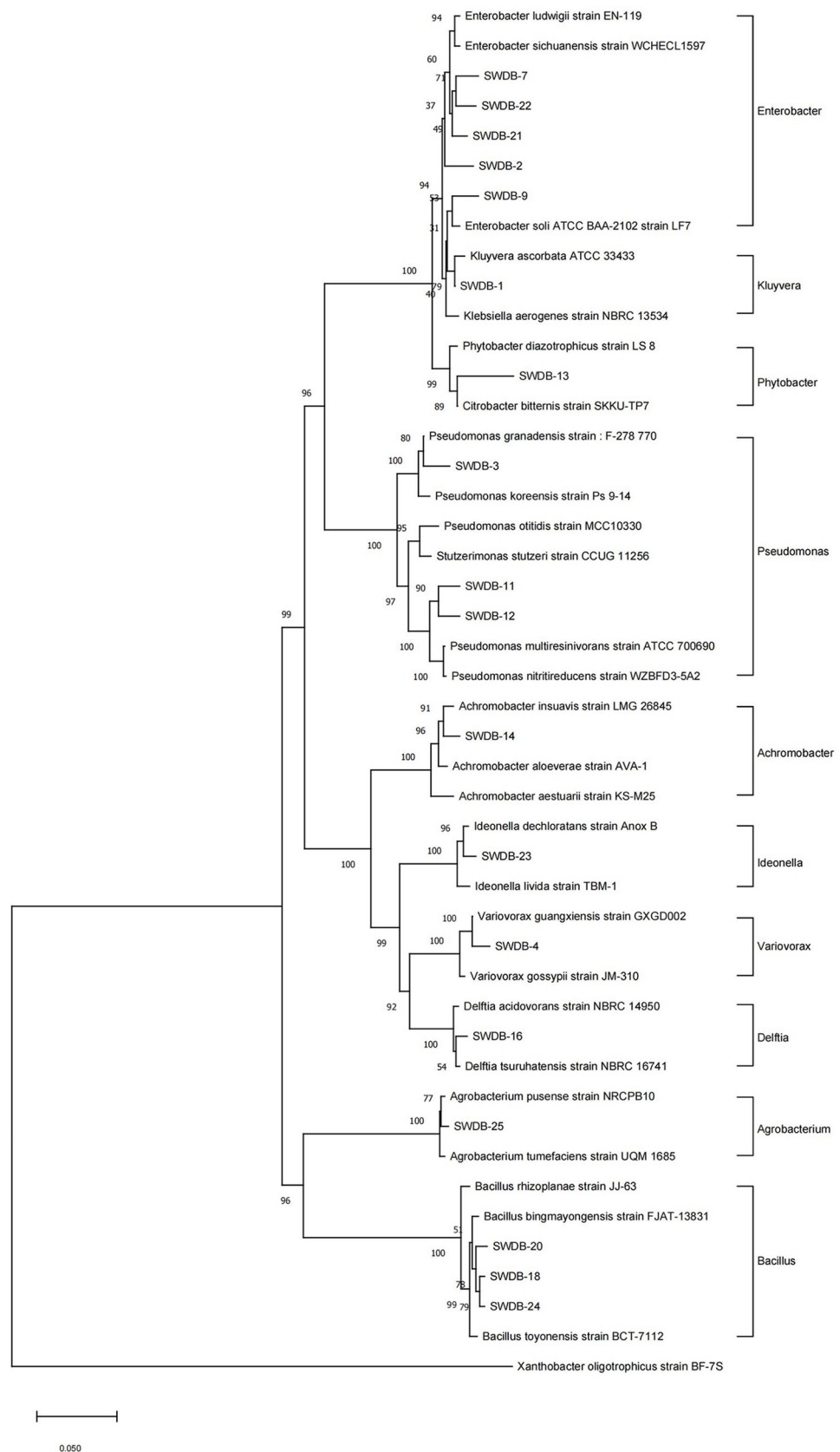

**Fig 1. Phylogenetic tree based on 16S rRNA gene sequence.**

hours. The ultraviolet and visible spectrophotometry experimental outcomes are presented in S2 Fig.

**3.3.2 Determination of degradation efficiency by HPLC.** The twenty-six strains were cultured in minimal salt liquid medium containing DBP with a concentration of 0.5 mmol/L for 48 h to form bacterial fermentation broth. The remaining concentration of DBP in the bacterial fermentation broth was determined by HPLC. Then the degradation efficiency was calculated by comparing it to the DBP standard curve. Table 2 lists the calculation results.

As shown in Table 2, the concentration of residual DBP in the bacterial fermentation broth was significantly reduced compared with that in the replicate, and the differences reached a

**Table 2. The time of growth, OD, residual DBP concentration, degradation efficiency, absorbance difference at maximum absorption wavelength and transparent circle.**

| Bacteria Strain | Cult-ure time (h)[i] | $OD_{600}$ in the bacterial fermentation broth[ii] | Residual DBP concentration in the bacterial fermentation broth (mmol/L)[iii] | Degradation efficiency in the bacterial fermentation broth (%) | absorbance difference at maximum absorption wavelength in the bacterial fermentation broth[iv] |
|---|---|---|---|---|---|
| CK | | | $0.332^a$ | $2.73\pm0.02^y$ | $1.75^v$ |
| SWDB-1 | 48h | $2.493\pm0.032^a$ | $0.098\pm0.004^{gh}$ | $71.25\pm0.01^l$ | 0.85 |
| SWDB-2 | 48h | $1.702\pm0.017^h$ | $0.250\pm0.007^c$ | $26.56\pm0.02^u$ | 0.59 |
| SWDB-3 | 48h | $2.075\pm0.029^{de}$ | $0.087\pm0.002^{hij}$ | $74.34\pm0.01^i$ | 0.94 |
| SWDB-4 | 48h | $1.836\pm0.010^g$ | $0.086\pm0.003^{hij}$ | $74.74\pm0.01^g$ | 0.96 |
| SWDB-5 | 48h | $1.833\pm0.015^g$ | $0.098\pm0.005^{gh}$ | $71.35\pm0.01^k$ | 0.86 |
| SWDB-6 | 48h | $1.986\pm0.023^f$ | $0.095\pm0.003^{ghi}$ | $72.03\pm0.01^j$ | 0.90 |
| SWDB-7 | 48h | $2.477\pm0.038^a$ | $0.074\pm0.004^{jk}$ | $78.17\pm0.01^d$ | 1.05 |
| SWDB-8 | 48h | $1.717\pm0.014^h$ | $0.082\pm0.002^{ijk}$ | $75.81\pm0.01^f$ | 1.09 |
| SWDB-9 | 48h | $1.131\pm0.023^l$ | $0.253\pm0.010^c$ | $25.82\pm0.03^v$ | 0.58 |
| SWDB-10 | 48h | $0.912\pm0.016^n$ | $0.217\pm0.011^e$ | $36.42\pm0.03^q$ | 0.74 |
| SWDB-11 | 48h | $2.483\pm0.031^a$ | $0.108\pm0.007^{fg}$ | $68.45\pm0.02^o$ | 0.81 |
| SWDB-12 | 48h | $2.365\pm0.027^c$ | $0.255\pm0.009^c$ | $25.17\pm0.03^w$ | 0.55 |
| SWDB-13 | 48h | $2.054\pm0.021^e$ | $0.080\pm0.003^{ijk}$ | $74.66\pm0.01^h$ | 0.95 |
| SWDB-14 | 48h | $2.112\pm0.018^d$ | $0.120\pm0.008^f$ | $64.75\pm0.02^p$ | 0.77 |
| SWDB-15 | 48h | $2.419\pm0.033^b$ | $0.028\pm0.005^l$ | $91.86\pm0.01^a$ | 1.20 |
| SWDB-16 | 48h | $0.984\pm0.015^m$ | $0.104\pm0.011^g$ | $69.60\pm0.03^n$ | 0.82 |
| SWDB-17 | 48h | $1.314\pm0.012^k$ | $0.293\pm0.006^b$ | $14.12\pm0.02^x$ | 0.47 |
| SWDB-18 | 48h | $1.573\pm0.011^i$ | $0.233\pm0.012^d$ | $31.78\pm0.04^r$ | 0.73 |
| SWDB-19 | 48h | $1.981\pm0.017^f$ | $0.248\pm0.011^{cd}$ | $27.24\pm0.03^t$ | 0.62 |
| SWDB-20 | 48h | $0.882\pm0.009^n$ | $0.068\pm0.005^k$ | $80.10\pm0.01^c$ | 1.00 |
| SWDB-21 | 48h | $2.051\pm0.026^e$ | $0.079\pm0.002^{jk}$ | $76.77\pm0.01^e$ | 1.11 |
| SWDB-22 | 48h | $1.589\pm0.018^i$ | $0.099\pm0.008^{gh}$ | $71.10\pm0.02^m$ | 0.84 |
| SWDB-23 | 48h | $1.546\pm0.015^i$ | $0.243\pm0.015^{cd}$ | $28.83\pm0.04^s$ | 0.63 |
| SWDB-24 | 48h | $1.434\pm0.022^j$ | $0.082\pm0.006^{hij}$ | $75.83\pm0.02^f$ | 1.10 |
| SWDB-25 | 48h | $0.777\pm0.013^o$ | $0.098\pm0.009^{gh}$ | $71.25\pm0.03^l$ | 0.85 |
| SWDB-26 | 48h | $0.730\pm0.009^p$ | $0.067\pm0.004^k$ | $80.26\pm0.01^b$ | 1.12 |

Notes: Replicate (CK): Noninoculated strain

[i]: Culture time: 48 h

[ii]: $OD_{600}$: The corresponding OD value of the strain was cultured in minimal salt liquid medium (containing 0.5 mmol /L DBP) for 48 h

[iii]: The remaining DBP concentration of the strain was cultured in minimal salt liquid medium (containing 0.5 mmol /L DBP) for 48 h. In the same column, different letters, such as a, b, and c, indicate significant differences ($P<0.05$), while the same letters indicate no significant differences ($P > 0.05$). Data containing letters such as ab were not significantly different from data containing a and b

[iv]: At the maximum absorption wavelength, the difference between the absorbance of each bacterial fermentation broth and that of the replicate

[v]: the average absorbance corresponding to the maximum absorption wavelength of the bacterial fermentation broth produced by the control.

significant level ($P < 0.05$). There were significant differences in the degradation capability of most DBP degrading bacteria ($P < 0.05$). After the twenty-six strains were cultured in minimal salt liquid medium for 48 hours with a DBP concentration of 0.5 mmol/L, their degradation capability of DBP was 0.048–0.313 mmol/L and degradation efficiency ranged from 14.12% to 91.86%. There were 7 strains with degradation efficiency above 75%, accounting for 26.92% of the total number of strains, and degradation capability was above 0.256 mmol/L. Among them, the degradation efficiency of SWDB-15 was the highest (91.86%), and the degradation capability of DBP was 0.313 mmol/L, which was significantly higher than that of other bacteria ($P<0.05$). The obvious difference in the degradation efficiency of DBP degrading bacteria may be due to the relatively simple culture conditions designed in this study. Different degrading bacteria have different optimal degradation conditions. Under the same culture conditions, the degradation capability of some degrading bacteria has not been fully demonstrated, resulting in lower degradation efficiency. Or different degrading bacteria have different functional genes and different degradation pathways, leading to differences in degradation efficiency. However, The specific reasons need to be further discussed and studied in the future.

## 4 Discussion

This study found that there were abundant and diverse DBP degrading bacteria in the vegetable field polluted by plastic mulch in Dayi County, Chengdu, Sichuan Province, China. The isolated strains were classified into two phyla and ten genera. Duan et al. [24] and Pan et al. [25] isolated and screened a strain of DBP degrading bacterium belonging to the *Pseudomonas* genus from vegetable fields of South China Agricultural University and facility vegetable fields of Qingdao, respectively. Yang et al. [32] screened a DBP degrading bacterium of the *Achromobacter* genus from the soil near a waste disposal station. Fang et al. [33] isolated an *Enterobacter* genus DBP degrading bacterium from a landfill bioreactor. Patil et al. [34] screened a strain of *Bacillus* genus DBP degrading bacterium from activated sludge. Wu et al. [35] isolated a DBP degrading bacterium belonging to the *Agrobacterium* genus from river sludge. Patil et al. [36] isolated and screened a *Delftia* genus DBP degrading bacterium from activated sludge. However, *Variovorax*, *Ideonella*, *Beijerinckia*, and *Phytobacter* isolated and screened in this study were rarely reported as DBP degrading bacteria.

According to the method that strains with a similarity of 16S rDNA sequence greater than 98.65% are classified as the same species [37], among the twenty-six strains of DBP degrading bacteria that have been screened. The 16S rDNA sequence similarity between SWDB-7 and *Enterobacter sichuanensis* was 97.58%. Therefore, SWDB-7 was initially identified as a potential new species. Although the use of 16S rDNA sequencing for taxonomic identification is one of the most effective and prevalently employed approaches, it can not only reveal the disparities among different bacterial genera but also facilitate obtaining their order through sequencing technology. Nevertheless, the use of 16S rDNA sequencing for taxonomic identification method does have limitations. Due to minor differences between species, some strains cannot be identified solely by 16S rDNA, and alternative identification methods are requisite. Hence, the polyphasic taxonomy of SWDB-7 can be carried out in future studies to identify which species SWDB-7 belongs and whether SWDB-7 is a potential new species.

This study found that DBP degrading bacteria with high degradation efficiency could be isolated from vegetable fields polluted by plastic mulch in Dayi County, Chengdu, Sichuan Province, China. Duan et al. [24], Pan et al. [25], Wang et al. [26] separated and screened a total of 4 strains of DBP degrading bacteria from agricultural soil in other areas, and their capability to degrade DBP is detailed in the "introduction" section. In addition, Duan [38] isolated a DBP degrading bacterium from PAEs contaminated soil. Under the condition that the

initial concentration of DBP in soil was 25 mg/kg, this strain reduced the DBP to 2.2 mg/kg within 28 days, and the degradation efficiency was 91.01%.

The degradation efficiency of SWDB-15 isolated in this study was 91.86%, which was higher than that of most of the DBP degrading bacteria isolated at present, but there was still a certain gap compared with the degradation efficiency of one of the bacteria isolated by Duan et al. [24] The possible reason is that the high initial concentration of DBP designed in this study led to low final degradation efficiency. The DBP degradation capability of SWDB-15 was 0.46 mmol/L, about 127.84 mg/L, while the DBP degrading bacteria isolated by Duan et al. [24] had a DBP degradation capability of 39.46 mg/L. It may also be because the culture conditions designed in this study were single, which did not reach the best degradation conditions for the degrading bacteria.

Although the five strains of DBP degrading bacteria SWDB-3, SWDB-10, SWDB-12, SWDB-16, and SWDB-19 did not produce clear transparent rings on minimal salt solid medium containing 0.5 mmol/L DBP. However, ultraviolet and visible spectrophotometry and HPLC determined that they still have the capability to degrade DBP. The reason may be that their capability to degrade is not strong enough, resulting in the transparent circle is not obvious. It may also be because different degrading bacteria have different degradation mechanisms, and the resulting degradation products are themselves opaque or insoluble in water. It is also possible that enzymes produced by some degrading bacteria do not spread. Therefore, strains that can produce transparent circles on culture plates must have degradation effects, and strains that do not have transparent circles on culture plates may also have degradation effects. The plate transparent ring method can screen out the strains with obvious transparent ring phenomenon, while the plate isolates without transparent ring phenomenon should be further screened by ultraviolet and visible spectrophotometry and HPLC. The results of ultraviolet and visible spectrophotometry and HPLC were analyzed to assess the degradation capability of DBP degrading bacteria. It was observed that a higher absorbance difference at the absorption peak, as measured by ultraviolet and visible spectrophotometry, corresponded to a greater degradation efficiency as determined by HPLC. Conversely, a lower absorbance difference indicated a lower degradation efficiency. However, few degrading bacteria exhibit high degradation efficiency but a low absorbance difference measured at the maximum absorption wavelength. For example, SWDB-20 demonstrates significantly greater degradation efficiency measured by HPLC compared to SWDB-8, SWDB-21, and SWDB-24. Nevertheless, the absorbance difference corresponding to the maximum absorption wavelength is smaller than that of SWDB-8, SWDB-21, and SWDB-24. The reason may be that the degradation mechanism of these degrading bacteria is different, and the degradation products produced are also different. The degradation of DBP includes two steps: side chain degradation and benzene ring degradation. Side chain degradation is the first step, including β-oxidation, transesterification, and degreasing. Among them, the degradation product of β-oxidation is diethyl phthalate (DEP) [39], the degradation product of transesterification is dimethyl phthalate (DMP) [40], and the degradation product of degreasing is monobutyl Phthalate (MBuP) [41–44]. The degradation of the benzene ring is an important step to realize the utilization of carbon source in DBP. DBP produces major metabolic intermediates polyamide (PA), protocatechuic acid (PCA), and n-butyl acetate (BAc) through reduction and decarboxylation. Under aerobic conditions, PA is converted into cis-3, 4-dihydro-3, 4-dihydroxyphthalate by Gram-positive bacteria and cis-4, 5-dihydro-4, 5-dihydroxyphthalate by Gram-negative bacteria [22, 44]. β-ketoadipic acid and 4-carboxyl-2-dihydroxy-muconic hemialdehyde can be produced by PCA when Ortho— or meta-cleavage ring occurs. After a series of metabolic processes, oxaloacetic acid and pyruvate produced will enter the tricarboxylic acid cycle (TAC) cycle [22, 42]. There are four typical degradation mechanisms of BAc, the first is the BAc→PCA metabolic path [22], the second is

the BAc→ 1-cyclohexene carboxylic acid → 2-hydroxy-cyclohexane carboxylic acid → adipic acid path [41], and the third is the BAc→ 2-hydroxybenzoic acid → phenol → catechol → cis-cis-adienedioic acid → miconolactone path [45, 46]. The fourth is BAc→ unstable intermediates (e.g., cis-1, 6-dihydroxy-2, 4-cyclohexadiene-1-carboxylic acid), which then enter a metabolic process similar to path 3 [22]. Ultraviolet and visible spectrophotometry does not separate DBP from degradation products, and degradation products may also absorb the maximum wavelength of light energy, resulting in reduced light intensity and resulting in smaller measurement results.

In summary, the plate transparent ring method offers the advantages of being fast and convenient for the preliminary screening of DBP degrading bacteria. However, further confirmation and assessment are still required using ultraviolet and visible spectrophotometry or/and HPLC. Compared with HPLC, ultraviolet and visible spectrophotometry is more user-friendly and requires simpler equipment. Nevertheless, its drawback lies in the determination of mixed degradation substances which may interfere with measurement reliability. Therefore, it can only serve as a preliminary method to assess the strength of DBP degrading bacteria's degradation capability. On the other hand, HPLC utilizes separation columns to accurately detect DBP in a reliable manner but involves heavy operation and a lengthy determination process. Each method has its own pros and cons; therefore, selection should be based on specific usage conditions.

This study provides a better theoretical basis for the treatment of DBP pollution caused by plastic mulch. The selected DBP degrading bacteria enrich the species resources for controlling DBP pollution in agricultural soil, and these DBP degrading bacteria can be used to develop bactericides to act on DBP-contaminated soil and reduce the DBP content in soil, which is of great significance for the bioreactor and treatment of DBP-contaminated soil.

The drawback of this study lies in that when investigating the degradation characteristics of DBP degrading bacteria, the culture time of the bacterial fermentation broth is set at 2 days and the culture conditions remain the same. However, in reality, the optimal degradation conditions and degradation time for each strain vary. And the maximum degradation capability of each strain cannot be ascertained by the uniform setting of culture conditions and culture time. This might lead to the neglect of some DBP degrading bacteria with superior degradation capability and excellent degradation efficiency. Therefore, in the subsequent research process, the optimal degradation conditions and degradation time of each strain can be taken into account to determine the maximum degradation efficiency of each strain, and identify the strain with the strongest degradation capability and the best degradation effect.

## 5 Conclusions

The diversity of DBP degrading bacteria was abundant in the plastic mulch polluted vegetable field in Dayi County, Chengdu City, Sichuan Province, China. The twenty-six strains belonged to two phyla, four classes, five orders, eight families, and ten genera, of which *Enterobacter* was the dominant genus. *Variovorax*, *Ideonella*, *Beijerinckia*, and *Phytobacter* were rarely reported as DBP degrading bacteria. SWDB-7 is a potential new species.

The DBP degrading bacteria in the plastic mulch polluted vegetable field in Dayi County, Chengdu City, Sichuan Province, China had strong degrading capability. Among the twenty-six strains of DBP degrading bacteria, the degradation efficiency of seven strains exceeded 75%, the degradation efficiency of strain SWDB-15 was the highest, the degradation efficiency was 91.86%, the degradation amount was 0.313 mmol/L, and the degradation efficiency of other strains was between 14.12% and 80.26%. The degradation amount was between 0.048 and 0.274 mmol/L.

In this study, the DBP degrading bacteria isolated and screened have rich diversity, stable and efficient degradation characteristics, and there are potential new species.

## Supporting information

**S1 Table. The online BLAST results of 16S rDNA sequences for the 26 DBP degrading bacteria strains in a vegetable field polluted by plastic mulch in Dayi County, Chengdu City, Sichuan Province, China.**
(XLSX)

**S1 Fig. Transparent circle images of some DBP degrading bacteria strains on minimal salt solid medium containing 0.5 mmol/L DBP.**
(TIF)

**S2 Fig. Ultraviolet and visible spectrophotometry spectrum of each DBP degrading bacteria strain.**
(ZIP)

## Author Contributions

**Conceptualization:** Yong Li.

**Data curation:** Sidan Gong.

**Formal analysis:** Yong Li.

**Investigation:** Huan Liu.

**Software:** Huan Liu.

**Writing – original draft:** Sidan Gong.

**Writing – review & editing:** Yong Li, Sidan Gong, Yajie Li.

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
