## [Decision Letter · Decision Letter 0]

16 Jun 2024

PONE-D-24-19447Screening and Diversity Analysis of Dibutyl Phthalate Degrading Bacteria in Agriculture SoilPLOS ONE

Dear Dr. Li,

Thank you for submitting your manuscript to PLOS ONE. After careful consideration, we feel that it has merit but does not fully meet PLOS ONE’s publication criteria as it currently stands. Therefore, we invite you to submit a revised version of the manuscript that addresses the points raised during the review process.

We look forward to receiving your revised manuscript.

Kind regards,

Trung Quang Nguyen

Academic Editor

PLOS ONE

Journal Requirements:

Additional Editor Comments:

I want to add some comments.

- The chemicals and materials need to be supplemented with a grade to indicate that they have the appropriate purity for research.

- Please present the QA/QC method to demonstrate the necessary parameters of the DBP analytical method, such as recovery efficiency, repeatability, and LOQ.

- The manuscript should illustrate more results using charts. For example, in section 3.3.1, a UV-Vis spectrum is required. Graphs showing the degradation of DBP over time should also be presented in chart form. Additionally, it may be necessary to include HPLC spectra to show the degradation before and after the degradation experiment.

---

Reviewers' comments:

Reviewer's Responses to Questions

**Comments to the Author**

1. Is the manuscript technically sound, and do the data support the conclusions?

Reviewer #1: Partly

Reviewer #2: Yes

Reviewer #3: Partly

Reviewer #4: Yes

Reviewer #5: Yes

2. Has the statistical analysis been performed appropriately and rigorously? 

Reviewer #1: N/A

Reviewer #2: N/A

Reviewer #3: No

Reviewer #4: Yes

Reviewer #5: Yes

3. Have the authors made all data underlying the findings in their manuscript fully available?

Reviewer #1: No

Reviewer #2: No

Reviewer #3: Yes

Reviewer #4: Yes

Reviewer #5: No

4. Is the manuscript presented in an intelligible fashion and written in standard English?

Reviewer #1: Yes

Reviewer #2: No

Reviewer #3: No

Reviewer #4: Yes

Reviewer #5: Yes

5. Review Comments to the Author

Reviewer #1: The article is well written and organized. Novelty is appreciated. Minor revisions are required.

1- Write the name of the country China for Dayi County. Also, for all cities in the abstract.

2- The following related articles should be cited in the introduction

https://doi.org/10.3390/ijms17071012

https://doi.org/10.1016/j.btre.2017.04.002

International Journal of Environmental Science and Development, Vol. 3, No. 3, June 2012

Biodegradation of Dimethyl Phthalate, Diethyl Phthalate, Dibutyl Phthalate and Their Mixture by Variovorax Sp.

https://doi.org/10.1016/j.scitotenv.2019.02.385

Li, H., Liu, L., Xu, Y. et al. Microplastic effects on soil system parameters: a meta-analysis study. Environ Sci Pollut Res 29, 11027–11038 (2022). https://doi.org/10.1007/s11356-021-18034-9

3- Novelty over old published articles should be highlighted in the abstract

4- Line 64- 66 add reference

5- Line 132, add country name. also, in line 136

6- Add reference for the method in line 153

7- Line 237, write full name for R2

8- Line 252, remove and

9- Line 269, as shown in Table 3-1.; it should be Table 1.also, in 271

10- Remove repeated information from line 350- 361

11- Figures for results of UV spectra and HPLC chromatograms should be displayed.

12- Study limitation and future plan should be suggested

Reviewer #2: The manuscript discussed the isolation and utilization of DBP bacterial degradation strains to reduce the DBP soil content. The bacterial strains were isolated using pure culture approach and were screened and identified using 16S rDNA sequencing. The isolated strains had a successful rate of DBP degradation, making them a good candidate for DBP removal from the soil with the strain SWDB-15 having the most effective degrading capability. However, a thorough English editing is required for this manuscript.

The researchers thoroughly discussed the background of the problem in the introduction. They discussed how DBP is related to plastic mulch and human health, and the governmental regulations pertaining to it in China and USA. They also compared its existence in vegetable fields compared to rice and other fields and thoroughly discussed previous literature pertaining to plastic degrading bacteria. Later, the authors discussed the methods thoroughly and showed how the DBP bacteria was identified and isolated. The authors also discussed the results of their work in details showcasing the abilities of the various bacterial strains in degradation of DBP and comparing their results to literature findings.

The work was done using standard methodology pertaining to such types of research and the results are adequately and clearly presented. However, several linguistic errors were found in the manuscript. In what follows, I will discuss those errors along with other comments I had:

1) Line 52, it is not usual that the phrase “in our nation” be used in scientific literature. Please replace it with the country name.

2) Line 75, instead of using the word “cheap”, you can say “cost-effective” or “relatively cheaper”

3) Line 111, replace on with in

4) Lines 126-132 are better suited at the beginning not towards the end of the introduction.

5) Line 135, no mention of the number of samples collected? These should be clarified

6) Line 137, should state that these are coordinates and not just numbers.

7) Line 149, the statement seems to be unrelated to text before it and should be integrated into it properly.

8) Line 154, same as above and additionally the first sentence is not clear here and the second sentence is in brackets which makes no sense for a sentence!

9) Line 158, it is better to use the word “Each” instead of “The” at the start of the sentence.

10) Line 168, “The bacterial suspension should be” I believe that it was already and thus this sentence should be re-written.

11) Line 172, the entire sentence needs to be re-written.

12) Line 177, the sentence “After that, inoculate the pure…..” needs to be re-written correctly.

13) Line 218, “then use sterile water….” should be re-written correctly.

14) Line 228, “place the conical…..” should be re-written correctly.

15) Line 236, what do the authors mean by “external standard method”? also, I would say “a standard curve was constructed” instead of construct a standard curve.

16) Line 242, the second sentence needs to be re-written.

17) Line 304, “Then the degradation efficiency can be calculated by comparing it to the DBP standard curve” you mean it can or it was?

18) Table 3-2: the letters should be put as superscripts instead of being next to the numbers.

19) In line 308, But the table doesn't exactly show which ones were in the bacterial fermentation broth? This distinction needs to be made clear in the table itself.

20) Line 345, replace the comma with a full stop.

21) Line 446, “the” should be capitalized correctly.

22) In addition, the statistical model used for the data analysis was not mentioned nor was the software used for the analysis which should typically be in the materials and methods section.

Reviewer #3: The techniques used, UV spectroscopy and HPLC and how the experiments and the results were presented is not convincing enough to agree that the Dibutyl Phthalate has been degraded by the bacteria. Also it will be interesting to know what the degraded components of Dibutyl Phthalate were using the bacteria isolated from this study. The bacteria could have degraded the Dibutyl Phthalate to other substance that can be regarded as pollutants.

The author made use of unclear (ambiguous) statements such as the ''using the pure culture approach'', ''purified by the dilution coating plate method and quadrant streaking method''. The experimental design of this research make some conclusions that can not to be affirmative. For one of the bacteria strains the author suggests it to be a new species based on 16S RNA blast only. This alone can not make a bacteria to be a new species. The author thinks a particular genera is abundant but based on the isolation and characterization technique used the abundance of that genera might not be the true picture.

Why did the author selected some colonies on the basal medium and not all. "Colonies with clear morphology and good growth but without transparent circle were separated and purified." I would have thought ability of the bacteria to grow on the DPB would have made them a presumptive bacteria capable of degrading DPB.

Not sure the need of purification in line 135

The soil could have been used without air drying not sure the use of air drying line 140

The unit should be rpm but rmp was used in the manuscript

10-1 bacterial suspension in line 167 could have 10 fold serial dilution

Not sure if a single gene can be assembled into contigs line 199

The isolated colonies in line 214 does not suggest each isolated bacteria

Both rpm and g was used in line 217. I suggest that a particular unit be used either g or rpm

I am confused with the word thallus in line 219

Was the UV reading at 200-350 a good wavelength for measurement of the breakdown of DPB

In the results section line 264 Table 2 was cited before table 1. The description of table 2 in line 261-264 is not what we have in the exact Table 2.

Table 3-1 might be moved to a supplementary file. Instead of having this table, it can be presented in a summary of presenting the gene length of the 16S RNA what range it is how many of the strains falls into what genus and a similarity range for the blast. A summary is ok and the table can then be moved into supplementary file.

Reviewer #4: Dear Authors,

The manuscript titled "Screening and Diversity Analysis of Dibutyl" presents an important investigation into the bioremediation potential of DBP-degrading bacteria in agricultural soil, which is a significant environmental and health concern. The research appears to be well-designed, and the findings contribute valuable insights into the microbial diversity and degradation capabilities of DBP in soil.

I have the following suggestions for the manuscript improvement:

1. It would be beneficial to include a brief overview of the current state of DBP pollution in China and globally to provide a broader context.

2. It would be helpful to include more details on the soil sampling process, such as the depth of sampling and the number of replicates, to ensure that the study's robustness is clear.

3. The use of 16S rDNA sequencing for taxonomic identification is appropriate, but the manuscript could benefit from a discussion on the limitations of this method and any additional techniques that could be used to confirm the identity of the potentially new species SWDB-7.

4. The manuscript could be strengthened by discussing the implications of the findings for practical applications, such as the potential for using these bacteria in field-scale bioremediation projects.

5. It could be strengthened by discussing the implications of these findings for future research or practical applications in bioremediation.

6. The abstract mentions “vegetable plots contaminating plastic mulch” which could be rephrased for clarity to “vegetable plots contaminated with plastic mulch”.

Overall, the manuscript is well-written and presents an important contribution to the field of environmental microbiology and bioremediation. With the suggested improvements, the manuscript could be further strengthened and provide even more value to the scientific community.

Reviewer #5: 1. The title needs to be clarified.

2. I highly recommend reviewing the grammar of the paper.

3. Introduction section;

- Line 54, change country exceeds to “countries exceed”.

- Line 56, write “Based”

- line 60, 77, 80, and 99, Change to agricultural soil.

- Line 66, remove “And”.

- Line 128, remove “so as”.

4. Results section;

- Where is the figure for Uv-Vis spectroscopy before and after degradation?

- Line 230, what is the meaning of 1.75?

- Change the table 3-2 to figure.

- The image of transparent rings on culture plates.

6. PLOS authors have the option to publish the peer review history of their article (what does this mean?). If published, this will include your full peer review and any attached files.

Reviewer #1: **Yes: **Mohammed Gamal

Reviewer #2: **Yes: **Mohammed Ali Al Abri

Reviewer #3: No

Reviewer #4: **Yes: **Dr. Niaz Ali

Reviewer #5: No

---

## [Author Response · Author response to Decision Letter 0]

27 Aug 2024

Comments and Suggestions for Journal Requirements:

Reply: We have revised the manuscript and file naming according to PLOS ONE's style requirements.

Reply: No permits were required for the sampling site. For details, see the attachment naming "a brief statement on the permits of the sampling site".

Comments and Suggestions for Editor:

1. The chemicals and materials need to be supplemented with a grade to indicate that they have the appropriate purity for research. 

Reply: Accepted and revised other chemical reagents’ grade in “Materials and Methods” section.

2. Please present the QA/QC method to demonstrate the necessary parameters of the DBP analytical method, such as recovery efficiency, repeatability, and LOQ.

Reply: Accepted and presented the relevant content in “Materials and Methods” section.

3. The manuscript should illustrate more results using charts. For example, in section 3.3.1, a UV-Vis spectrum is required. Graphs showing the degradation of DBP over time should also be presented in chart form. Additionally, it may be necessary to include HPLC spectra to show the degradation before and after the degradation experiment.

Reply: The images of ultraviolet and visible spectrophotometry spectrum have been supplemented. However, wing to the malfunction of the computer that stored figures for results of HPLC chromatograms in our laboratory, the relevant data has been lost. Currently, we are unable to provide the relevant figures and we sincerely apologize for that.

Comments and Suggestions for Reviewer #1: 

1- Write the name of the country China for Dayi County. Also, for all cities in the abstract.

Reply: Accepted and it is revised.

2- The following related articles should be cited in the introduction.

Reply: Accepted and it is revised. However, this article (https://doi.org/10.1016/j.scitotenv.2019.02.385) has already been cited as reference [20] in the introduction.

3- Novelty over old published articles should be highlighted in the abstract.

Reply: Accepted and it is revised in “Introduction” section . 

4- Line 64- 66 add reference

Reply: Accepted and it is revised.

5- Line 132, add country name. also, in line 136

Reply: Accepted and it is revised. 

6- Add reference for the method in line 153

Reply: Accepted and it is revised.

7- Line 237, write full name for R2

Reply: Accepted and it has been improved. 

8- Line 252, remove and

Reply: Accepted and it is removed. 

9- Line 269, as shown in Table 3-1.; it should be Table 1.also, in 271

Reply: Accepted and it is revised.

10- Remove repeated information from line 350- 361

Reply: Accepted and it is removed.

11- Study limitation and future plan should be suggested

Reply: Accepted and it has been improved. 

Reviewer #2: 

1) Line 52, it is not usual that the phrase “in our nation” be used in scientific literature. Please replace it with the country name.

Reply: Accepted and it has been changed into “in China”. 

2) Line 75, instead of using the word “cheap”, you can say “cost-effective” or “relatively cheaper”

Reply: Accepted and it is corrected.

3) Line 111, replace on with in

Reply: Accepted and it is corrected.

4) Lines 126-132 are better suited at the beginning not towards the end of the introduction.

Reply: Accepted and it is revised.

5) Line 135, no mention of the number of samples collected? These should be clarified

The number of samples collected was not mentioned in the previous manuscript. It has been improved and revised in the "Materials and Methods" section.

6) Line 137, should state that these are coordinates and not just numbers.

Reply: Accepted and it is revised.

7) Line 149, the statement seems to be unrelated to text before it and should be integrated into it properly.

Reply: Accepted and it is revised.

8) Line 154, same as above and additionally the first sentence is not clear here and the second sentence is in brackets which makes no sense for a sentence!

Reply: Accepted and it is revised.

9) Line 158, it is better to use the word “Each” instead of “The” at the start of the sentence.

Reply: Accepted and it is revised.

10) Line 168, “The bacterial suspension should be” I believe that it was already and thus this sentence should be re-written.

Reply: Accepted and this sentence has been re-written.

11) Line 172, the entire sentence needs to be re-written.

Reply: Accepted and the entire sentence has been re-written.

12) Line 177, the sentence “After that, inoculate the pure…..” needs to be re-written correctly.

Reply: Accepted and the sentence has been re-written.

13) Line 218, “then use sterile water….” should be re-written correctly.

Reply: Accepted and this sentence has been re-written.

14) Line 228, “place the conical…..” should be re-written correctly.

Reply: Accepted and this sentence has been re-written.

15) Line 236, what do the authors mean by “external standard method”? also, I would say “a standard curve was constructed” instead of construct a standard curve.

External standard method is one of the common methods of instrument analysis, and it is a kind of comparative method. First, a certain amount of standard product is added in a blank solvent according to the gradient to make a reference sample. Then, measuring the peak areas corresponding to the different reference samples. After that, based on the resulting peak area and the corresponding sample concentration, plot the standard curve. Finally, measuring the peak area of the sample with unknown concentration and combining the resulting standard curve calculating the concentration of the sample with unknown concentration. 

About “construct a standard curve”, it is corrected, changing into “a standard curve was constructed”. 

16) Line 242, the second sentence needs to be re-written.

Reply: Accepted and this sentence has been re-written.

17) Line 304, “Then the degradation efficiency can be calculated by comparing it to the DBP standard curve” you mean it can or it was?

Yes, it can. The specific calculation method and process are detailed in the interpretation of the "external standard method".

18) Table 3-2: the letters should be put as superscripts instead of being next to the numbers.

Reply: Accepted and it is revised.

19) In line 308, But the table doesn't exactly show which ones were in the bacterial fermentation broth? This distinction needs to be made clear in the table itself.

The OD600, Residual DBP concentration, Degradation efficiency and absorbance difference at maximum absorption wavelength were in the bacterial fermentation broth. It has been refined and modified in the manuscript.

20) Line 345, replace the comma with a full stop.

Reply: Accepted and it is corrected.

21) Line 446, “the” should be capitalized correctly.

Reply: Accepted and it is corrected.

22) In addition, the statistical model used for the data analysis was not mentioned nor was the software used for the analysis which should typically be in the materials and methods section.

Reply: Add “statistical analysis” in the “materials and methods” section.

Reviewer #3: 

1.It will be interesting to know what the degraded components of Dibutyl Phthalate were using the bacteria isolated from this study. The bacteria could have degraded the Dibutyl Phthalate to other substance that can be regarded as pollutants.

In previous manuscript’ "Discussion" section, relevant content regarding the degradation mechanism of DBP degrading bacteria has already been presented. In future experiments, we will conduct further research on the degradation mechanism of DBP degrading bacteria which we had separated and gain an understanding of the degradation products of each DBP degrading bacterium.

2.The author made use of unclear (ambiguous) statements such as the ''using the pure culture approach'', ''purified by the dilution coating plate method and quadrant streaking method''. The experimental design of this research make some conclusions that can not to be affirmative. For one of the bacteria strains the author suggests it to be a new species based on 16S RNA blast only. This alone can not make a bacteria to be a new species. The author thinks a particular genera is abundant but based on the isolation and characterization technique used the abundance of that genera might not be the true picture. 

About unclear (ambiguous) statements and inconclusive conclusions, we have revised the article according to the Reviewer’ comment. About SWDB-7, we only thought that it is a potential new species. In future studies, the polyphasic taxonomy of SWDB-7 would be carried out in future studies to identify which species SWDB-7 belongs and whether SWDB-7 is a potential new species. And about prominent genus, we have revised the article according to the Reviewer’ comment.

3.Why did the author selected some colonies on the basal medium and not all. "Colonies with clear morphology and good growth but without transparent circle were separated and purified." I would have thought ability of the bacteria to grow on the DPB would have made them a presumptive bacteria capable of degrading DPB.

 Since we hold the belief that the bacteria in colonies with clear morphology or good growth but without transparent circle possess a superior degradation ability on minimal salt solid medium. Then, use ultraviolet and visible spectrophotometry and HPLC to validate their degradation capability.

4.Not sure the need of purification in line 135

Reply: Accepted and it is removed. 

5.The soil could have been used without air drying not sure the use of air drying line 140

Reply: Accepted and sampling description had been revised.

6.The unit should be rpm but rmp was used in the manuscript

Reply: Accepted and it is corrected.

7.10-1 bacterial suspension in line 167 could have 10 fold serial dilution

Reply: Accepted and it is revised.

8.Not sure if a single gene can be assembled into contigs line 199

Reply: A single gene can’t be assembled into contigs. The relative description has been revised.

9.The isolated colonies in line 214 does not suggest each isolated bacteria

Reply: Accepted and the relative description has been revised. 

10.Both rpm and g was used in line 217. I suggest that a particular unit be used either g or rpm

Reply: Accepted and it is corrected.

11.I am confused with the word thallus in line 219

Reply: The relative description has been revised.

12.Was the UV reading at 200-350 a good wavelength for measurement of the breakdown of DPB

Yes, it was. By reading the relevant literature and materials, such as: Lu MY. (2019) Isolation of PAEs-degrading strain、analysis of its degradation pathway, cloning of hydrolase gene and characterization of its encodinig enzyme. M.Sc. Thesis, Nanjing Agricultural University. Xia FY. (2002) Biodegradability Research of Phthalic Acid Esters. D.Eng. Thesis, Zhejiang University. and so on, we found that the scanning range of 200-350 nm can contain the measurement wavelength required for DBP.

13.In the results section line 264 Table 2 was cited before table 1. The description of table 2 in line 261-264 is not what we have in the exact Table 2.

Reply: Accepted and it is revised.

14.Table 3-1 might be moved to a supplementary file. Instead of having this table, it can be presented in a summary of presenting the gene length of the 16S RNA what range it is how many of the strains falls into what genus and a similarity range for the blast. A summary is ok and the table can then be moved into supplementary file.

Reply: Accepted and it is revised.

Reviewer #4: 

1. It would be beneficial to include a brief overview of the current state of DBP pollution in China and globally to provide a broader context.

Reply: Accepted and it is revised in “Introduction” section.

2. It would be helpful to include more details on the soil sampling process, such as the depth of sampling and the number of replicates, to ensure that the study's robustness is clear.

Reply: Accepted and it is revised in “Materials and Methods” section.

3. The use of 16S rDNA sequencing for taxonomic identification is appropriate, but the manuscript could benefit from a discussion on the limitations of this method and any additional techniques that could be used to confirm the identity of the potentially new species SWDB-7.

Reply: Accepted and it is revised in “Discussion” section.

4. The manuscript could be strengthened by discussing the implications of the findings for practical applications, such as the potential for using these bacteria in field-scale bioremediation projects.

Reply: Accepted and it is revised in “Discussion” section.

5. It could be strengthened by discussing the implications of these findings for future research or practical applications in bioremediation.

Reply: Accepted and it is revised in “Discussion” section.

6. The abstract mentions “vegetable plots contaminating plastic mulch” which could be rephrased for clarity to “vegetable plots contaminated with plastic mulch”.

Reply: Accepted and it is corrected. 

Reviewer #5: 

1. The title needs to be clarified.

Reply: Accepted and the title has been improved.

2. I highly recommend reviewing the grammar of the paper.

Reply: Accepted and the grammar of the paper has been revised.

3. Introduction section;

1)- Line 54, change country exceeds to “countries exceed”.

Reply: Accepted and it is corrected. 

2)- Line 56, write “Based”

Reply: Accepted and it is corrected. 

3)- line 60, 77, 80, and 99, Change to agricultural soil.

Reply: Accepted and it is corrected. 

4)- Line 66, remove “And”.

Reply: Accepted and it is removed.

5)- Line 128, remove “so as”.

Reply: Accepted and it is removed.

4. Results section;

1)- Line 230, what is the meaning of 1.75?

Reply: 1.75 means that the average absorbance corresponding to the maximum absorption wavelength of the bacterial fermentation broth produced by the control. The relative description has been revised.

2)- The image of transparent rings on culture plates.

Reply: We have added the image of transparent rings on culture plates as S1 Fig. in “Supporting information” section.

---

## [Decision Letter · Decision Letter 1]

11 Sep 2024

Screening and Diversity Analysis of Dibutyl Phthalate Degrading Bacteria in Agricultural Soil in Chengdu, China

PONE-D-24-19447R1

Dear Dr. Yong Li,

We’re pleased to inform you that your manuscript has been judged scientifically suitable for publication and will be formally accepted for publication once it meets all outstanding technical requirements.

Kind regards,

Trung Quang Nguyen

Academic Editor

PLOS ONE

Additional Editor Comments (optional):

Reviewers' comments:

Reviewer's Responses to Questions

**Comments to the Author**

1. If the authors have adequately addressed your comments raised in a previous round of review and you feel that this manuscript is now acceptable for publication, you may indicate that here to bypass the “Comments to the Author” section, enter your conflict of interest statement in the “Confidential to Editor” section, and submit your "Accept" recommendation.

Reviewer #1: All comments have been addressed

Reviewer #3: All comments have been addressed

Reviewer #4: All comments have been addressed

Reviewer #5: All comments have been addressed

2. Is the manuscript technically sound, and do the data support the conclusions?

Reviewer #1: Yes

Reviewer #3: Yes

Reviewer #4: Yes

Reviewer #5: Yes

3. Has the statistical analysis been performed appropriately and rigorously? 

Reviewer #1: N/A

Reviewer #3: I Don't Know

Reviewer #4: Yes

Reviewer #5: Yes

4. Have the authors made all data underlying the findings in their manuscript fully available?

Reviewer #1: Yes

Reviewer #3: Yes

Reviewer #4: Yes

Reviewer #5: Yes

5. Is the manuscript presented in an intelligible fashion and written in standard English?

Reviewer #1: Yes

Reviewer #3: Yes

Reviewer #4: Yes

Reviewer #5: Yes

6. Review Comments to the Author

Reviewer #1: The authors did all required recommendations. I appreciate their responses. The paper could be published in the current form.

Reviewer #3: (No Response)

Reviewer #4: Dear Authors,

Thank you for addressing the suggestions. The manuscript has improved significantly. However, please correct “agriculture” to “Agriculture” in Line #51.

Best regards,

Dr. Niaz Ali

Reviewer #5: (No Response)

7. PLOS authors have the option to publish the peer review history of their article (what does this mean?). If published, this will include your full peer review and any attached files.

Reviewer #1: **Yes: **Mohammed Gamal

Reviewer #3: No

Reviewer #4: **Yes: **Dr. Niaz Ali

Reviewer #5: **Yes: **Shaimaa Alexeree

---

## [Editor Report · Acceptance letter]

23 Sep 2024

PONE-D-24-19447R1 

PLOS ONE

Dear Dr. Li, 

I'm pleased to inform you that your manuscript has been deemed suitable for publication in PLOS ONE. Congratulations! Your manuscript is now being handed over to our production team.

Kind regards, 

on behalf of

Dr. Trung Quang Nguyen 

Academic Editor

PLOS ONE